



# The ice-nucleating activity of African mineral dust in the Caribbean boundary layer

Alexander D. Harrison[1], Daniel O'Sullivan[1], Michael P. Adams[1], Grace C. E. Porter[1], Edmund Blades[2], Cherise Brathwaite[3], Rebecca Chewitt-Lucas[3], Cassandra Gaston[4], Rachel Hawker[1], Ovid O. Krüger[5], Leslie Neve[1], Mira L. Pöhlker[5, 6, 7], Christopher Pöhlker[5], Ulrich Pöschl[5], Alberto Sanchez-Marroquin[1], Andrea Sealy[3], Peter Sealy[8], Mark D. Tarn[1], Shanice Whitehall[3], James B. McQuaid[1], Kenneth S. Carslaw[1], Joseph M. Prospero[4] and Benjamin. J. Murray[1]

[1] School of Earth and Environment, University of Leeds, Woodhouse Lane, Leeds, England
[2] Queen Elizabeth Hospital, Martindales Road, St. Michael, Barbados
[3] Caribbean Institute for Meteorology and Hydrology, Bridgetown, Barbados
[4] Rosenstiel School of Marine and Atmospheric Science, University of Miami, Miami, U.S.A
[5] Multiphase Chemistry Department, Max Planck Institute for Chemistry, Mainz, Germany
[6] Leibniz Institute for Tropospheric Research (TROPOS), Leipzig, Germany
[7] Institute for Meteorology, University of Leipzig, Stephanstraße, Leipzig, Germany
[8] AEROCE Research Facility, Ragged Point, St Philip, Barbados

*Correspondence to*: Benjamin J. Murray (b.j.murray@leeds.ac.uk)

**Abstract.** African mineral dust is transported many thousands of kilometres from its source regions and, because of its ability to nucleate ice, it plays a major role in cloud glaciation around the globe. The ice-nucleating activity of desert dust is influenced by its mineralogy, which varies substantially between source regions and across particle sizes. However, in models it is often assumed that the activity (expressed as active sites per unit surface area as a function of temperature) of atmospheric mineral dust is the same everywhere on the globe. Here, we find that the ice-nucleating activity of African desert dust sampled in the summertime marine boundary layer of Barbados (July and August, 2017) is substantially lower than parameterizations based on soil from specific locations in the Saharan desert or dust sedimented from dust storms. We conclude that the activity of dust in Barbados' boundary layer is primarily defined by the low K-feldspar content of the dust, which is around 1%. We propose that the dust we sampled in the Caribbean was from a region in West Africa (in and around the Sahel in Mauritania and Mali), which has a much lower feldspar content than other African sources across the Sahara and Sahel.

## 1. Introduction

The formation and growth of ice crystals strongly affects the properties of clouds and has important implications for cloud-climate feedbacks [Storelvmo, 2017]. In the absence of a special class of aerosol particle that can catalyse ice formation, known as ice-nucleating particles (INPs), cloud droplets can supercool to temperatures below −33 °C before freezing homogenously [Herbert *et al.*, 2015; Rosenfeld and Woodley, 2000]. INPs can result in ice nucleation in droplets at much warmer temperatures, triggering microphysical processes that can substantially alter the development of cloud systems. For example, shallow clouds can persist in a supercooled state over large parts of the world, but can transition to a lower albedo state if INPs



are present [Murray *et al.*, 2021; Storelvmo *et al.*, 2015; Tan *et al.*, 2016; Vergara-Temprado *et al.*, 2018]. In deep convective
clouds the presence of INPs, ingested from the boundary layer or entrained from higher levels, can strongly alter the extent
and lifetime of anvil cirrus through perturbation of the microphysics and dynamics in the mixed-phase region of the cloud
[Gibbons *et al.*, 2018; Hawker *et al.*, 2021b; Rosenfeld *et al.*, 2011]. Despite the importance of INPs for clouds and climate,
the global distribution, seasonal cycles, transport, sources and sinks of INPs remain poorly characterised [Kanji *et al.*, 2017;
Vergara-Temprado *et al.*, 2017].

Mineral dust from the world's deserts is one of the most important INP types for mixed-phase clouds around the globe, even
in locations many thousands of kilometres from the arid source regions [Burrows *et al.*, 2013; Hoose *et al.*, 2010; Vergara-
Temprado *et al.*, 2017; Wiacek *et al.*, 2010]. It is often assumed that the activity of mineral dust in the atmosphere does not
vary with source or physical and chemical processing in the atmosphere [DeMott *et al.*, 2015; Niemand *et al.*, 2012; Ullrich *et
al.*, 2017; Zhao *et al.*, 2021]. Also, some commonly used laboratory derived parameterisations for the ice-nucleating activity
of mineral dust are based on the ice-nucleating activity of desert dust samples collected from the surface in source regions
rather than material transported 100s or 1000s of kilometres through the atmosphere [Connolly *et al.*, 2009; Niemand *et al.*,
2012; Ullrich *et al.*, 2017]. However, there are reasons why these parameterisations may not represent the ice-nucleating
activity of all atmospheric mineral dust from deserts: firstly, dusts from different source regions, with different mineralogies
[Claquin *et al.*, 1999; Nickovic *et al.*, 2012] have been found in some studies to have substantial differences in activity [Boose
*et al.*, 2019; Boose *et al.*, 2016b]; secondly, dust may be physically or chemically altered (referred to as weathering or aging)
when transported in the atmosphere [Fahy *et al.*, In press; Sullivan *et al.*, 2010].

Mineral dusts generated from desert soils are composed of various minerals, with the three most abundant groups in the
atmosphere being clay, quartz and feldspar [Murray *et al.*, 2012; Perlwitz *et al.*, 2015]. These dusts are created by the physical
and chemical weathering of the Earth's surface and are commonly sourced from arid regions in Africa, the Middle East and
Asia [Prospero *et al.*, 2002a]. Feldspar minerals rich in potassium (K-feldspars) are thought to be the most effective ice
nucleator of any major mineral component found in dust [Atkinson *et al.*, 2013; Augustin-Bauditz *et al.*, 2014; Harrison *et al.*,
2019; Harrison *et al.*, 2016; Holden *et al.*, 2019; Niedermeier *et al.*, 2015; Peckhaus *et al.*, 2016; Zolles *et al.*, 2015]. The
amount of feldspar in desert soils varies substantially, for example in parts of the western Sahel in Mali and Mauritania the
total feldspar content is less than 2%, whereas in large parts of the central Sahara total feldspar content is ~12 to 20% [Nickovic
*et al.*, 2012; Perlwitz *et al.*, 2015]. Hence, it would be expected that the ice-nucleating activity of mineral dust has a dependence
on source region. In addition, there is significant variability in the ice-nucleating activity of K-feldspars with similar crystal
structures and chemical composition, which produces a range of ice-nucleating activities [Harrison *et al.*, 2019; Harrison *et
al.*, 2016].

The concentration of INPs in a dust plume will of course decrease as the plume mixes with non-dusty air, but the dust's activity
(active sites per unit surface area) may also change. For example, the ice-nucleating activity of dust has been shown to decrease





when exposed to acids [Kumar *et al.*, 2018; Perkins *et al.*, 2020; Sullivan *et al.*, 2010; Wex *et al.*, 2014]. This reduced activity is presumably related to the acid-dissolution of ice-active sites, but interaction with less reactive materials can also alter the nucleating activity of mineral dust. For example, laboratory studies show that internal mixing with a range of aqueous salts may reduce (or sometimes enhance) the activity of mineral dust [Boose *et al.*, 2019; Kumar *et al.*, 2018; Reischel and Vali,

1975; Whale *et al.*, 2018]. In addition, it has been reported that sea salt is correlated with reduced activity of desert dust in atmospheric samples [Iwata and Matsuki, 2018; Si *et al.*, 2019]. Conversely, it has been shown that ammonium salts can enhance nucleation [Reischel, 1987; Whale *et al.*, 2018]. Given the global significance of desert dust as an INP type, it is clearly necessary to characterise the ice nucleating ability of desert dust far from source regions as well as dusts samples close to those sources.In this paper, we report measurements of INP concentrations and ice-nucleating activity (on a per surface area

basis) of mineral dust from African sources in the marine boundary layer (MBL) on the east (windward) coast of Barbados during July and August 2017. Apart from offering insight into the activity of mineral dust advected from Africa, these measurements help to define the INP spectrum in the tropical Atlantic and the Caribbean boundary layer, which is important for defining primary ice production in deep convection. The summertime atmosphere above Barbados is well known for containing substantial quantities of dust which has been transported from Africa, typically over the course of about a week

[Prospero and Carlson, 1972; Prospero *et al.*, 2021; Zuidema *et al.*, 2019]. It is tempting to think of mineral dust from North Africa to the Caribbean as a simple source, transport and receptor system, however, dust transport across the Atlantic to the boundary layer of Barbados is complex. Hence, we start this paper with a brief review of dust transport that then informs our interpretation of the measurements we report later in the paper.

## 2.    Transport of African dust to the Caribbean

North Africa is the world's largest single source of mineral dust [Ginoux *et al.*, 2012; Prospero *et al.*, 2002b], exporting between 400 to 2200 Tg yr$^{-1}$ [Huneeus *et al.*, 2011]. Dust is transported across the Atlantic at a range of altitudes and the vertical distribution of dust is defined by the meteorology in the source regions. The well know Saharan air layer (SAL) is an important dust transport route above the boundary layer, but there are other transport routes which may be as or more important for the Barbados boundary layer (see Figure 1).

The SAL is a layer of hot dry air that is frequently dusty, which propagates westward towards the Caribbean [Carlson and Prospero, 1972; Weinzierl *et al.*, 2017]. Its origins are across the central Sahara desert, where air is heated and then mixed vertically upwards due to dry convection before sliding over the marine boundary layer (MBL) as it travels westward over the Atlantic [Carlson and Prospero, 1972; Knippertz and Todd, 2012]. The SAL is typically confined to ~4 km thickness, with dust concentrations decreasing sharply at its top boundary and a strong inversion at the bottom of the layer separating it from

the MBL [Carlson and Prospero, 1972; Gasteiger *et al.*, 2017; Weinzierl *et al.*, 2017]. Mid-level clouds can form in the upper part of the SAL where the dust may promote primary ice formation [Barreto *et al.*, 2022].



In the summer, the SAL usually rides above the MBL in the Caribbean, but the air in the MBL is also often very dusty [Maring *et al.*, 2003a; Reid *et al.*, 2002; Weinzierl *et al.*, 2017]. Some dust in the MBL comes from downward transport from the SAL, but there is also a low altitude transport route through the MBL of dusty air from western Africa [Carlson and Prospero, 1972; Chiapello *et al.*, 1995; Reid *et al.*, 2002]. Radiogenic isotope signatures of dust sampled in the Barbados MBL suggest that the dominant source of transported dust is in the western Sahel around Mali and Senegal rather than sources in the central Sahara, such as the Bodélé depression [Bozlaker *et al.*, 2018; Pourmand *et al.*, 2014].

Observations show very little or no detectable sea salt aerosol above the boundary layer, indicating that turbulent mixing between the MBL and free troposphere is weak [Maring *et al.*, 2003a]. Hence, it is not a surprise that dust concentrations in the MBL and SAL are often decoupled. In fact, the dust concentrations in the MBL can be greater than those in the SAL [Maring *et al.*, 2003a; Reid *et al.*, 2002], which is consistent with independent transport routes and African sources. In addition, the size distribution of dust in the SAL is similar to the size distribution in the MBL on average [Maring *et al.*, 2003a], which suggests that size-dependent sedimentation of SAL dust into the MBL is not a major process and that dust properties are dependent on transport history.

It typically takes around a week for African mineral dust to reach the Caribbean once leaving the west coast of Africa [Knippertz and Todd, 2012; Weinzierl *et al.*, 2017]. The highest concentrations of dust often occur behind a tropical easterly wave (a trough of low pressure) and because the SAL is relatively warm, convection is suppressed in regions of high dust [Carlson and Prospero, 1972; Prospero and Carlson, 1970]. Instead, relatively shallow clouds tend to form at the top of the MBL and deep convection is associated with air masses containing less dust [Prospero and Carlson, 1970]. Maring *et al.* [2003a] point out that wet scavenging was a minor loss route, since the small cumulus clouds which form in dusty air do not regularly precipitate. Also, these small cumulus clouds are not supercooled and INPs are therefore not preferentially scavenged through growth and precipitation of ice.

The size distribution of African dust is remarkably stable across the Atlantic. Based on gravitational settling alone, one would expect the coarse mode aerosol to be depleted on transport across the Atlantic. However, the normalised size distribution of particles less then ~7 µm diameter is similar on both sides of the Atlantic, but with particles larger than 7 µm being preferentially removed [Maring *et al.*, 2003b]. Modelling indicates that other processes including turbulent mixing in the SAL play a role in counteracting gravitational settling [Gasteiger *et al.*, 2017; Maring *et al.*, 2003b].

Overall, the literature paints a complex picture of the relationship between the properties of dust measured in Barbados and dust transport from Africa. However, it is clear that the properties of dust in the Caribbean MBL may be distinct from those in the SAL since the sources and transport pathways are distinct.



## 3. Methodology

### 3.1 Project overview

The Barbados Ice-nucleating particle Concentration Experiment (B-ICE) was conducted at Ragged Point (13°09'55.6"N,
59°25'54.7"W) which is on the far easterly windward coast of Barbados and has been used to study African dust for many decades [Prospero *et al.*, 2021]. The air at this location is relatively pristine with little local influence and consistently originates from the east [Zuidema *et al.*, 2019]. The University of Miami has infrastructure on-site including a ~17 m sampling tower, which is situated on a 30 m bluff, and was built to minimize the sampling of local sea spray aerosol produced at the base of the cliff [Prospero *et al.*, 2021]. It has been shown that the aerosol sampled here are representative of the marine boundary
layer [Wex *et al.*, 2016]. The summer months are when Barbados typically experiences the highest dust concentrations [Zuidema *et al.*, 2019], and so B-ICE ran during these months (July 24th to August 24th, 2017).

Our approach in B-ICE was to sample aerosol onto filters for offline analysis of INP concentration spectra. INP droplet freezing analysis was performed on-site at Ragged Point within the IcePod mobile laboratory within hours of sampling [O'Sullivan *et al.*, 2018]. This minimised any changes in ice-nucleating activity caused by storage and transport of filter samples back to our
laboratory in Leeds (UK). Using the IcePod to analyse the samples also had the advantage that we could perform successive handling blanks in order to identify and minimise sources of contamination. Given the relatively low INP concentrations we observed and the potential for contamination, the ability to do handling blanks on-site proved to be very important. In parallel with the INP analysis, we quantified the particle size distribution. The combination of these two measurements allowed us to derive the temperature-dependent active site density of dust in Barbados. The active site density allowed us to quantitatively
compare the ice-nucleating activity of African dust in the Barbados MBL with the activity of dust sampled from the surface and in other dusty locations.

### 3.2 Aerosol sampling and measurements

Three Mesa Labs BGI PQ100 aerosol samplers, atop the sampling tower, were used at a flow rate of 16.7 L min$^{-1}$ (i.e. 1 m$^3$ h$^{-1}$) with an aerosol cut-off size of 10 μm to sample aerosol onto track-etched polycarbonate filters (Whatman Nuclepore, 0.4
micron pore size). The samplers were allowed to run for varying periods of time with the flow rate and sampling duration being recorded by inbuilt mass flow controllers (see Table S1). Often, two PQ100 samplers were run side-by-side to obtain two filters that could be directly compared. Technical problems with the pumps meant that sampling periods were not always uniform, but the reported sample volumes are accurate. Where possible, one filter would be used for the on-site ice nucleation experiments (section 2.3) and the other would be sealed, frozen and shipped to Leeds for analysis by scanning electron
microscopy with energy-dispersive X-ray spectroscopy (SEM-EDS, Tescan VEGA3 XM SEM fitted with an X-max 150 SDD EDS system controlled by an Aztec 3.3 software by Oxford Instruments) (section 3). The methodology for SEM-EDS analysis to provide details on aerosol mineralogy and particle size distributions is described by Sanchez-Marroquin *et al.* [2019]. In



addition, we used powder X-ray diffraction (XRD) to examine the mineralogy of a dust sample using a Bruker D8 instrument operating with Cu K-alpha 1 radiation.

Throughout the campaign, handling blanks were taken by placing a clean filter in the filter holder of a BGI PQ100 sampler, then removing it without sampling any aerosol through it and performing ice nucleation experiments following the procedure we used for analysing aerosol samples (see section 2.3). For some additional handling blanks, a high-efficiency particulate air (HEPA) filter was used instead of the inlet head and air was sampled for a duration of time before removing the filter for analysis.

A TSI 3321 Aerodynamic Particle Sizer (APS) was placed in a weatherproof container on top of the tower with no size-selective inlet head. Non-dried air was sampled at a flow rate of 1 L min$^{-1}$ to take particle size distribution measurements in the range of 0.5-20 μm. Maring et al. (2003) suggest that a change in relative humidity from 25% to 95% will lead to a <6% change in geometric diameter, with the typical relative humidity at Barbados being ~80%. Hence, hygroscopic growth will have a minor impact on the resulting size distributions. The size distribution of smaller aerosol particles (~10 nm – 500 nm, volume equivalent diameter) was quantified using a GRIMM 5420 Scanning Mobility Particle Sizer (SMPS). The flow was dried to around 30% RH and had a sheath flow in the inlet at the top of the tower to achieve near isokinetic sampling. The SMPS was situated in a laboratory just to the side of the tower and was connected to an inlet with an almost vertically oriented pipe.

The ambient aerosol size distribution was measured throughout the field study by both the SMPS and APS, with the exception

of periods affected by sporadic power outages or maintenance. SMPS and APS datasets were converted to volume equivalent diameters and merged to produce the volume equivalent size distribution, as described by Mohler *et al.* [2008], using a particle density of 2.6 g cm$^{-3}$ and a shape correction factor of 1.3 (i.e. values pertinent for mineral dust).

### 3.3 Ice nucleation experiments

An immersion mode droplet assay technique, the microlitre Nucleation by Immersed Particle Instrument (μL-NIPI), was used for all ice nucleation experiments [Harrison *et al.*, 2016; O'Sullivan *et al.*, 2014; Whale *et al.*, 2015]. This method has previously been applied to the analysis of filter-collected aerosol samples [O'Sullivan *et al.*, 2018], based on the method of filter sampling and subsequent suspension of the aerosol particles in water described by DeMott *et al.* [2016]. In brief, aerosol particles were washed from polycarbonate filters using ultrapure water by placing each filter in a 50 mL Falcon tube with 5

mL of water and agitating on a shaker for 30 min. The resulting wash water was then used to pipette 1 μL droplets (roughly 30-50 droplets per experiment) onto a hydrophobic glass slide which was atop a cold stage and sealed from the atmosphere in a Perspex enclosure. Dry nitrogen gas was passed into the chamber to prevent condensation on to the glass slide while the system was cooled down at 1 ˚C min$^{-1}$. The temperature at which the droplets froze was recorded using a digital camera and





synchronised thermocouple temperature measurements. This leads to the recording of a fraction of droplets frozen at a given

temperature for each aerosol suspension. The fraction frozen can then be used to calculate the atmospheric INP concentration $N_{INP}(T)$; where the square brackets indicate concentration) and the active site density per unit surface area ($n_s(T)$) of the aerosol suspension, see equations 1 and 2.

$$N_{INP}(T) = -\ln\left(\frac{n_u(T)}{N}\right)\left(\frac{V_w}{V_a V_s}\right) \qquad (1)$$

$$n_s(T) = \frac{-\ln\left(1 - \frac{\Delta N(T)}{N}\right)}{A} \qquad (2)$$

where $n_u$ is the number of unfrozen droplets out of the total number of droplets ($N$), $V_w$ is the volume of wash water used to collect the aerosol from the filters (5 mL), $V_a$ is the volume of the droplets (0.001 mL), $V_s$ is the volume of air sampled on to the filters, $A$ is the surface area of aerosol particles in each droplet and $\Delta N$ is the number of frozen droplets at temperature $T$.

At the start of each day, a blank experiment was conducted by performing the droplet assay technique using high performance

liquid chromatography (HPLC) grade water to determine the background of the experiment before performing experiments on filter-collected aerosol. HPLC wash water from the rinsing of handling blank filters was also periodically assayed to determine the extent of contamination during the sampling process. The handling blanks and baselines were in agreement with one another, although there was a large degree of variability across all blanks (~4 °C at $T_{50}$, where $T_{50}$ is the temperature at which 50% of the droplets have frozen). This variability was attributed to contamination of the HPLC water, possibly from aerosol

present in the working environment. Efforts were made to minimise contamination, but backgrounds throughout the campaign were variable. These backgrounds were used to background subtract our data and can be seen in SI Figure S2 (also see text S1 for a full description of the protocol).

A subset of samples underwent heat-testing to evaluate the presence of protein-based biological INPs, since ice-active proteins associated with some classes of biological INPs can be denatured with heat, causing a decrease in their ice-nucleating activity

[Christner *et al.*, 2008; Garcia *et al.*, 2012; O'Sullivan *et al.*, 2018]. In contrast, it is assumed that the ice-nucleating activity of the most active component of mineral dusts, K-feldspar, is not heat sensitive, consistent with recent tests [Daily *et al.*, 2021]. A selection of suspensions were heated to 100 °C for 1 h by immersing a suspension in a sealed polypropylene falcon tube within boiling water. The INP content of this suspension was determined before and after heating.

The INP data presented in the following sections has had the pure water background subtracted (unless specified). Data points

that had error bars spanning more than four orders of magnitude, or were considered statistically insignificant, are indicated with open symbols and should be considered as upper limits. A description of the background subtraction and the calculation of the errors is found in text S1.



### 4. Characterisation of aerosol in Barbados during the campaign

A compilation of size distribution measurements at Ragged Point using the SMPS and APS are shown in Figure 2. These

distributions show that the aerosol particle size distribution at Ragged Point during the campaign were relatively invariant and always contained the same modes centred at about 0.04, 0.15 and 1.5 µm. The mode at 0.15 µm is probably associated with sea spray, whereas the 0.04 µm was probably associated with new particle formation events [Quinn *et al.*, 2021; Wex *et al.*, 2016]. The mode at 1.5 µm is associated with mineral dust, which is similar to the mode centred at 2.0 µm associated with mineral dust in the Amazon [Moran-Zuloaga *et al.*, 2018]. This invariance in the measured size distributions during the

campaign is consistent with a relatively constant source of mineral dust and subsequent transport.

The SEM-EDS-derived size-resolved composition of aerosol samples collected at Ragged Point are shown in Figure 3. This data shows that mineral dust was the dominant aerosol type in the accumulation and coarse mode (above ~300-600 nm) in this location. Mineral dust is identified as Si rich material, which includes materials like quartz, silica, clays, and feldspars, as well as a smaller proportion of Ca rich material. This analysis suggests that mineral dust was the main contributor to the surface

area of the aerosol in this location with, on average, 92% of the aerosol surface area being attributable to mineral dust above 0.2 µm. We use this observation later in this paper to justify the assumption that the aerosol surface area derived from the APS and SMPS size distribution measurements is a good approximation of the mineral dust surface area in the derivation of $n_s(T)$.

There are also other particle types contributing to the aerosol in Barbados. The mineral dust was often internally mixed with sea salt, similar to a previous study of aerosol at Ragged Point [Kandler *et al.*, 2018], and sea salt (Na rich category) also

appeared as an externally mixed (mostly submicron) particle type. Previous work has linked sea salt aerosol to wind speed [Kandler *et al.*, 2018; Klingebiel *et al.*, 2019].

Using the SEM-EDS analysis, we also quantified the size distribution of the aerosol and compared this to the volume equivalent size distributions from the APS-SMPS system in Figure 2. We use these size distributions to define $n_s(T)$, the density of active sites – an expression of active sites per unit surface area. The aerosol surface area determined from these two very different

methods of deriving size distributions is within a factor of ~2. However, the size distribution from the SEM-EDS is shifted to larger sizes. This might be because particles on a filter will tend to orientate with the shorter dimension in the vertical, hence the true volume-equivalent diameter may be somewhat smaller. In addition, particles of mineral dust are often aggregates of smaller particles and may have a lower density than used to derive the volume equivalent diameter from the APS and SMPS data (we used 2.6 g cm$^{-3}$). It is also possible that we undersampled the coarse mode aerosol at the APS inlet, which would

result in a low bias in the surface area. When calculating the values of $n_s(T)$ for mineral dust in Barbados, presented in Section 5.2, we need the surface area of mineral dust. We integrated over the full size distribution, however aerosol types other than mineral dust will likely dominate below ~200 nm, hence we will overestimate the mineral dust surface area by a factor of much less than two. Nevertheless, the uncertainty in mineral dust surface area is less than a factor of two overall, which is relatively small in comparison to other uncertainties in the derivation of $n_s(T)$ (see Text S1 in the SI).



Given K-feldspars have been identified as the mineral group that dominates the ice-nucleating activity of mineral dusts, an estimate of the K-feldspar content of the dust is therefore useful. Kandler *et al.* [2018] reported that pure K-feldspar grains made up 0.7 ±0.2 wt% of the mineral dust (across ~1 to 10 µm diameter; average taken from data in their Fig. 13) sampled at Ragged Point (summer 2013 and 2016). Using the method for analysing SEM-EDS spectra defined by Kandler *et al.* [2018] suggests that 0.1 wt% of mineral dust grains were made of K-feldspar in our samples. This value is likely an underestimation

of the K-feldspar component due to the technique not being sensitive to the detection of K-feldspar when it is internally mixed with other minerals [Kandler *et al.*, 2018]. The K-feldspar content of dust in other locations around the world can be substantially larger than this, ranging from values comparable to those reported here up to 25% in Morocco [Atkinson *et al.*, 2013]

X-ray diffraction was used to analyse dust accumulated in the rain water trap on a BGI PQ100 sampler (between the 03/08/17-

05/08/17). This was an opportunistic sampling during a heavy rain event and was not done routinely during the campaign. The Rietveld analysis of the resulting powder pattern (Figure S4) revealed that there was kaolinite (57.8 wt%), muscovite and/or illite) (27 wt%), quartz (9.1 wt%), plagioclase (3.6 wt%) and calcite (2.6 wt%) present (Figure S4). K-feldspar was below the detection limit of the method, which we estimated at ~2 wt% [Hillier, 1999; Maters *et al.*, 2019]. Overall, K-feldspar being in concentrations below 2 wt% and above 0.1 wt% is consistent with previous measurements at Barbados of 1.7 wt% made by

Glaccum and Prospero [1980] and ~0.7 wt% by Kandler *et al.* [2018].

## 5. Ice-nucleating particle measurements in Barbados

### 5.1 Ice-nucleating particle concentrations in Barbados

The INP concentrations measured in the MBL at Ragged Point in Barbados during July and August 2017 are shown in Figure

3a. The concentrations are generally low, with between ~0.01 and 0.4 INP L$^{-1}$ at –20 °C. As a result of the low INP concentrations, the fraction frozen curves were close to the fraction frozen curves of the handling blanks, hence some of the data is marked as an upper limit (open symbols; see text S1 for details of the background subtraction). The slopes of the fraction frozen curves (dln$N_{INP}$/d$T$) were very similar throughout the campaign, indicating that the same INP type was present throughout. The aerosol (and dust) mass concentrations were relatively constant during the campaign (Figure 5), as were the

meteorological parameters of pressure, relative humidity and temperature (see Figure S7). The mineral dust concentration was 23 ± 15 µg m$^{-3}$ for much of the campaign (based on the APS data), with an excursion to concentrations of around 100 µg m$^{-3}$ around the 18$^{th}$ August. The time series in Figure 5b indicates that there is no obvious corresponding increase in INP concentration with this period of enhanced mineral dust. However, we would only expect a shift of ~2 °C for an increase of dust surface area of a factor of five, provided nothing else (such as dust mineralogy) changed.



To determine whether biological ice-nucleating material contributes to the INP population, we heated sample suspensions to about 100 °C. Protein-based biological INPs are denatured by heat and hence if they are present we expect to see a decrease in activity, whereas the activity of K-feldspar is affected only marginally (a deactivation of around 1 °C) [Daily *et al.*, 2021; O'Sullivan *et al.*, 2018]. On heating a selection of aerosol suspensions, we observed that there was no substantial decrease in ice-nucleating activity and therefore conclude that proteinaceous biological INPs were not the dominant INP type (Figure 6

and S8). However, inspection of the individual fraction frozen curves (Figure S8) shows that, on two occasions (2$^{nd}$ and 3$^{rd}$ August), the INPs active at the highest temperatures were sensitive to heat. These days were characterised by relatively high wind speeds and on the 3$^{rd}$ August the SEM-EDS analysis (Figure 2) revealed more coarse mode sea spray than other days. Hence, we suggest that, on some occasions, marine organic INPs were contributing to the INP population. Nevertheless, overall, the heat tests were consistent with mineral dust (specifically K-feldspar) being the major INP type at this location with

some possible sporadic contributions from biogenic (most likely marine organic) sources.

In Figure 4b, we compare the INP measurements with literature INP concentration data for the Caribbean region and a compilation of concentrations around the world from precipitation samples. The new data for Ragged Point in Barbados is consistent with other measurements in the MBL in other parts of the Caribbean [DeMott *et al.*, 2016; DeMott *et al.*, 2015; Schrod *et al.*, 2020]. Relative to the compilation of INP concentrations from Petters and Wright [2015], the INP concentrations

in the MBL of the Caribbean are at the low end of what is observed around the world.

In Figure 4c we also compare to measurements of INP concentrations in the eastern tropical Atlantic, much closer to the African dust sources. Price et al. (2018) reported aircraft measurements made in the eastern tropical Atlantic at altitudes between 30 m and 3.5 km, where INP concentrations were between ~1 and 100 L$^{-1}$ at –20 °C, whereas they were between 0.01 and 0.4 L$^{-1}$ in Barbados. Welti *et al.* [2018] reported ground based MBL measurements in Cape Verde (~10$^{-4}$ to 10$^{-2}$ L$^{-1}$ at -

10°C). Note that the technique employed by Welti et al. was insensitive to INP concentrations above about 0.1 L$^{-1}$, hence the actual INP concentrations may have extended to greater values at temperatures below about –15 °C. These INP concentrations overlap with those found in the Caribbean, but extend to higher values. Generally, the comparison of INP concentrations on the two sides of the Atlantic suggests that there is a lower INP concentration in the Caribbean versus the eastern tropical Atlantic. Qualitatively, this is what we would expect, given the surface area concentration of dust is lower in the Caribbean

than near the coast of Africa.

In an aircraft study, DeMott *et al.* [2015] reported that the INP concentrations in the MBL layer were considerably lower than in the free troposphere (Figure 4b). Both the MBL and the free troposphere are relatively dusty in the Caribbean, hence it would appear that the activity of the dust in the MBL and free troposphere is perhaps different, which might be consistent with different transport pathways or distinct aging processes in the two layers.






### 5.2 Ice-nucleating activity of African dust in Barbados

To quantify the ice-nucleating activity of the dust we sampled in the MBL of Barbados, we derive the temperature-dependent active sites per unit surface area, $n_s(T)$, for our aerosol samples. We derive $n_s(T)$ using the measured INP concentrations and the surface area of aerosol derived from the SMPS and APS data (Figure 2). We assumed that the surface area derived from the total size distribution measurements was equal to the surface area of mineral dust. This assumption is supported by the SEM-EDS analysis, which showed that mineral dust made up >90 % of the aerosol surface area of particles larger than 0.2 μm, the size range with the majority of the surface area (Figure 3).

In Figure 7, we present $n_s(T)$ determined from our measurements in Barbados as well as those reported in the literature for African mineral dust (we have focused this comparison exclusively on dust sourced from Africa, since the dust we sampled in Barbados originates from Africa). The literature data falls into two broad groups: the first is for African surface samples (soil or settled from a dust storm) that were taken to a laboratory for analysis of their ice-nucleating properties; the second is for samples of airborne African mineral dusts that were sampled and then analysed either in the field or in a laboratory.

The active site density, $n_s(T)$, of dust sampled in Barbados is about two orders of magnitude smaller than African dust examined in a cloud chamber that was sampled from specific locations [Ullrich *et al.*, 2017] and about one order of magnitude smaller than dust sampled in the eastern tropical Atlantic by Price *et al.* [2018]. The activity of African dust in Barbados overlaps with the activity of airborne African dust sampled in Israel [Reicher *et al.*, 2018], the Canary Islands [Boose *et al.*, 2016a], and several locations near or in Africa [Boose *et al.*, 2016b]. Our results show that transported mineral dust in the Caribbean MBL has a relatively low activity compared to dust samples in the African sources, hence parameterisations based on the ice-nucleating activity of surface samples from those sources will produce INP concentrations that are too high.

While the activity of dust in the MBL is relatively low, there is evidence that the activity of the dust in the Caribbean SAL is much greater. DeMott *et al.* [2015] found that the INP concentration in the SAL layer were higher than in the MBL during the ICE-T campaign in July 2011. They then used a dust parameterisation (based on a combination of lab and field measurements) to predict the INP concentration in the MBL and SAL using the measured aerosol size distribution. They found that the parameterisation represents the activity of mineral dust in the SAL, it overpredicts the activity of dust in the MBL by a factor of ~15. Hence, the question is why the African dust in the MBL of Barbados has a relatively low activity compared to dust in the SAL, the eastern tropical Atlantic and the African source .

### 6. Why African dust in the MBL in Barbados has a lower ice-nucleating activity compared to literature parameterisations and dust in other locations

The $n_s(T)$ values presented in Figure 7 and the discussion above show that dust in the MBL of Barbados has a lower activity than dusts used in laboratory experiments, or dust in the Caribbean SAL and the eastern tropical Atlantic. In this section we discuss the possible reasons why dust in Barbados has a relatively low ice-nucleating activity.





### 6.1. Low K-feldspar content

Mineralogy, and in particular the amount of K-feldspar, strongly determines the ice-nucleating activity of mineral dust [Atkinson *et al.*, 2013; Augustin-Bauditz *et al.*, 2014; Harrison *et al.*, 2019; Niedermeier *et al.*, 2015]. The mineralogy

measurements summarised in section 4 indicate that the K-feldspar content of dust in Barbados is typically around 0.1 to 2 % by mass. Hence, in Figure 7, we have plotted an $n_s(T)$ parameterisation from Harrison *et al.* [2019] for K-feldspar scaled to 1%, i.e., assuming 1% of the surface area of African dust is K-feldspar. Harrison et al. [2019] derived this parameterisation from data for K-feldspar from multiple samples, which we use to indicate the variability in $n_s(T)$ for K-feldspar as a shaded area in Figure 7. The 1% K-feldspar parameterisation is an excellent fit to the African dust in Barbados. It is also a good fit to

the $n_s(T)$ values derived from dust sampled in Israel [Reicher *et al.*, 2018], the Canary islands [Boose *et al.*, 2016a], and airborne dust sampled from several locations near or in Africa [Boose *et al.*, 2016b].

The $n_s(T)$ values for surface samples of African dust that were then aerosolised in a laboratory [Boose *et al.*, 2016b; Ullrich *et al.*, 2017], are around two orders of magnitude higher than those for Barbados, but with a shallower slope (see Figure 7). Some of the samples used by Ullrich *et al.* [2017] were particles that had sedimented over Egypt from dust storms that originated

over the Sahara, while others were direct samples of soil in Egypt (50 km north of Cairo), Morocco and Tunisia. These samples may be richer in coarser material and it is the coarser material that is known to contain more feldspar, compared to the finer fractions [Perlwitz *et al.*, 2015], and perhaps this is why they have a greater activity than airborne dust. Also, given the mineralogical diversity of African dust source regions [Nickovic *et al.*, 2012], dust samples from relatively few locations have been studied for their ice-nucleating activity. It is possible that sources relevant for Barbados in Western Africa, that have not

been directly tested for their ice-nucleating activity, have a lower activity than the samples included in the parameterisation defined by Ullrich *et al.* [2017].

There are a several potential reasons why the K-feldspar content of dust is lower in the MBL of Barbados than in the eastern tropical Atlantic or in the SAL layer. The first is that the source region of dust arriving in the Barbados MBL may be different to the source region of dust in the SAL or in the eastern tropical Atlantic. As discussed in section 2, there is a body of evidence

suggesting that the dominant transport route for African dust into the Caribbean MBL is from western Africa, whereas the SAL is sourced from further east in Africa. We examined back trajectories and they generally support the idea that low-level air is derived from coastal regions, but air in the SAL is derived from further east, but have low confidence in them due to the complex and poorly represented convective processes in western Africa. Mineralogy maps of surface soils show that there is a large area in West Africa with a very low total feldspar content (0-2 %), whereas large parts of the Sahara have 12-16 %

feldspar (this is all feldspars of which K-feldspars is one group [Nickovic *et al.*, 2012]). Nickovic *et al.* [2012] indicates that the very low feldspar region is in and around the Sahel in Mauritania and Mali, a region that has been implicated as a dust source relevant for the MBL of Barbados by isotope studies [Bozlaker *et al.*, 2018; Pourmand *et al.*, 2014]. Hence, our



observations in the MBL of Barbados of a low ice-nucleating activity for mineral dust are consistent with K-feldspar concentrations found from specific sources in western Africa that have been associated with transport to the Caribbean.

There may also be some chemical and physical processing during transport that reduces the K-feldspar content. K-feldspars are more abundant in the coarse mode than the accumulation mode [Perlwitz *et al.*, 2015] and since atmospheric lifetime decreases with particle size, the K-feldspar proportion and activity of dust is therefore expected to decrease during transport [Atkinson *et al.*, 2013]. Measurements indicate that only particles larger than about 7 µm diameter are lost through gravitational sorting across the Atlantic [Maring *et al.*, 2003b]. Size-resolved mineralogy measurements focusing on K-feldspar on both

sides of the Atlantic (in the same air mass) would be needed to resolve the question of how important loss of the largest particles is for the overall ice-nucleating activity of dust.

Chemical processing by internal mixing with reactive materials may also preferentially remove K-feldspar. Dust particles can become internally mixed with sea salt (Figure 2) and become immersed in a concentrated brine [Zhu *et al.*, 1992]. In addition, dust particles can become internally mixed with acids, such as sulphuric or nitric acid. When exposed to acids, feldspars react

and transform to clays. In several laboratory experiments the activity of mineral dust decreased on exposure to atmospherically relevant acids [Augustin-Bauditz *et al.*, 2014; Kumar *et al.*, 2018; Perkins *et al.*, 2020; Sullivan *et al.*, 2010; Wex *et al.*, 2014]. However, simulating the conditions a dust particle experiences in the atmosphere is challenging and the quantitative experiments we would need to describe the temperature and concentration dependence of the reactions have not been performed. Hence, we are unable to say how important this effect may be. However, for these reactions to uniformly reduce

the K-feldspar content to around 1% seems unlikely given atmospheric variability in transport and chemistry.

### 6.2. Mixing with sea salt aerosol

It has been observed that immersion of K-feldspar particles in dilute salt solutions reduces their ice-nucleating activity relative to immersion in pure water [Whale *et al.*, 2018]. This deactivation is not a colligative (i.e. thermodynamic) effect, but instead Na and Cl ions alter the kinetics of nucleation. Whale *et al.* [2018] observed a 1-2 °C decrease in activity for a $1.5 \times 10^{-4}$ M

NaCl solution, which increased to a 3-4 °C deactivation with $1.5 \times 10^{-3}$ M solution. Given that the mass concentrations of sea salt at Ragged Point typically range from 10 to 30 µg m$^{-3}$ during July and August [Stevens *et al.*, 2016], we would anticipate our dust suspensions made from the filter samples would have between $1.3 \times 10^{-4}$ and $3.8 \times 10^{-4}$ M NaCl. Hence, the mixing of salt with mineral dust might be expected to reduce freezing temperatures by around 1-3°C (which would correspond to a decrease in activity of up to about one order of magnitude). In future, efforts should be made to separate the salt and mineral

dust from filter samples in order to investigate the role of salt further. Also, the role of salt and mixing state in nucleation requires further attention since it is not clear what the effect of sea salt would be on ice nucleation when an aerosol particle activates to a cloud droplet [Whale *et al.*, 2018].

In summary, we conclude that the primary reason for the low ice-nucleating activity of dust in the MBL of Barbados, when compared to literature parameterisations, is that the K-feldspar content is less than 2%. There may also be other processes that





reduce the activity of transported dust including gravitational settling, chemical aging (weathering) and in particular interaction with sea salt.

### 7. Comparison of the measurements in Barbados with a global aerosol model

In Figure 8 we compare the INP concentrations predicted by the GLOMAP global aerosol model, described by [Vergara-
Temprado *et al.*, 2017], with the Barbados measurements. In GLOMAP, we represent the ice-nucleating activity of mineral dust with K-feldspar, for which we have a separate tracer in the model and take into account dust source mineralogy defined by Nickovic *et al.* [2012]. Previously, we used a parameterisation for the ice-nucleating activity of K-feldspar from Atkinson *et al.* [2013], but here we have instead used the more recent parameterisation from Harrison *et al.* [2019] that produces $n_s$ values a factor of 5-10 smaller between -15 and -25°C. For sea spray INPs, the other INP type that may be important in this
marine location, we model the emission of primary marine organic material and use the parameterisation defined by Wilson *et al.* [2015] to define the associated INP concentration.

The comparison in Figure 8 reveals that the slope of the INP concentration ($\mathrm{d}\log N_{\mathrm{INP}}/\mathrm{d}T$) is consistent with K-feldspar. However, the model-simulated $N_{\mathrm{INP}}$ values are higher by about a factor of five. Marine organics make a much smaller contribution to the INP concentration than mineral dust below about -20°C, but are predicted to make the dominant contribution
above about -15°C, where the INP concentration are always less than ~$10^{-2}$ L$^{-1}$. This is consistent with the results of our heat tests where marine organic INP are expected to be deactivated with heat, but where we observed no clear deactivation with heating in the bulk of our samples (see section 5.1).

The overestimation of INP concentrations by the model appears to be a result of the modelled K-feldspar proportion being too high. We reach this conclusion because the average aerosol mass predicted in the GLOMAP simulations during the campaign
was typically within a factor of two of observations (Figure 5), whereas GLOMAP predicted ~7 wt% of the total aerosol mass was K-feldspar, with ~8 wt% of the mineral dust mass component being attributed to K-feldspar (Figure. S10). In contrast, the XRD and SEM-EDS analyses from this study (see section 3) and the work of previous studies in the region [Glaccum and Prospero, 1980; Kandler *et al.*, 2018] showed the K-feldspar concentration to be in the range of 0.1 to 2 wt%, which suggests that the GLOMAP K-feldspar content is too high. Hence, it seems that GLOMAP predicts too much K-feldspar in the region
and if this were corrected the agreement would be very good.

There are several possible reasons why GLOMAP predicts too high a K-feldspar content of dust. One is that the meteorology near West Africa associated with the complex transport from the source regions is poorly represented in GLOMAP. This could give rise to a poor vertical structure of dust over the tropical Atlantic even though the surface mass concentrations of mineral dust in Barbados are well represented [Hawker, 2021]. Hence, the model may be mixing dust from high feldspar areas
in the central Sahara with those in western Africa with much lower amounts of feldspar.



Another hypothesis is that the size of K-feldspar-containing particles is not adequately represented in the model, even though the total dust mass concentration is consistent with observations in Barbados. Wet and dry size-dependent removal processes will preferentially remove larger particles, therefore the assumed distribution of the K-feldspar across the particle size distribution may critically influence the transport of K-feldspar. Perlwitz *et al.* [2015] suggest that Atkinson et al. (2013) places

too large a proportion of K-feldspar in the fine mode relative to the coarse mode, which results in too much transportation of K-feldspar. In GLOMAP, the size distribution of K-feldspar is based on the mineralogical maps presented in Nickovic *et al.* [2012] that provides the coarse (silt) mode feldspar content, but not the fine (clay) mode. The ratio of K-feldspar in the fine mode to the coarse mode is assumed to be the same ratio as for quartz (which is reported by Nickovic *et al.* [2012]). This assumption likely overpredicts the amount of K-feldspar in the clay mode, as feldspar is more readily broken down via

chemical reactions than quartz, which would lead to a diminishing population of feldspar particles in the clay mode relative to the more chemically stable quartz. Hence, the amount of K-feldspar that undergoes long-range transport will be dependent on the size distribution of K-feldspar at source, a property of the particles that is unfortunately poorly defined [Perlwitz *et al.*, 2015]. In opposition to our hypothesis, Perlwitz *et al.* [2015] suggest that there is more feldspar in the fine fraction than defined in Nickovic *et al.* [2012], which would mean size sorting is a less effective way of reducing K-feldspar content.

**8.   Conclusions**

African desert dust is dispersed around the globe and contributes to the INP population even many thousands of kilometres from the source regions across the Sahara and Sahel. Hence, a quantitative understanding of the ice-nucleating activity of African dust in the atmosphere is needed. We report measurements of INP concentrations in the summertime marine boundary layer of Barbados (July and August, 2017). INP concentrations during our campaign were comparable to other remote marine

locations around the world and much lower than many terrestrial influenced air masses. This was despite the Caribbean being known as a location influenced by African mineral dust.

The activity of African dust in Barbados, in terms of active sites per unit surface area, is several orders of magnitude lower than parameterisations based on dusts sampled from the surface in Africa that were later analysed in a laboratory. While there are several reasons why we might expect mineral dust that has been transported thousands of kilometres to have a lower ice-

nucleating activity than dust samples taken directly from the African source regions, it appears that the low activity is mainly a result of the dust mineralogy in Barbados. The activity of dust in Barbados is consistent with measurements of the K-feldspar content of dust sampled at Ragged Point in Barbados, of around 1%, which is lower than that measured in many African source regions [Atkinson *et al.*, 2013]. Previous dust isotope work suggests that dust arriving in the MBL of Barbados is preferentially from sources in West Africa [Bozlaker *et al.*, 2018; Pourmand *et al.*, 2014]. These regions, in and around the Sahel in

Mauritania and Mali, are known to have much lower feldspar contents than most of the rest of the African dust source regions [Nickovic *et al.*, 2012]. Hence, our measured dust activity is consistent with these isotope studies that suggest transport of dust to the MBL of Barbados from these specific regions in western Africa. Furthermore, previous work suggests that the activity



of dust in the Saharan air layer may be much greater than in the MBL [DeMott *et al.*, 2015], which is consistent with sources further east in Africa with greater feldspar content.

Comparison of our measurements with a global aerosol model (GLOMAP), reveals that the model over predicts the INP concentration in Barbados by about a factor of five. This high bias in the model is related to the K-feldspar content of the simulated dust arriving in Barbados being too high. While we represent the mineralogy of dust sources across Africa in the aerosol model, it seems that there are errors in either (or both) the transport and mixing of dust from source regions or the representation of the mineralogy of the dust at source.

The model correctly predicts that at concentrations above ~0.01 $L^{-1}$ , mineral dust from Africa is the dominant INP type rather than marine organics associated with sea spray. At lower concentrations and at temperatures above about –15 °C, the model suggests that marine organics become more important than mineral dust. This may mean that marine organics are important for primary ice production in Caribbean (and across the tropical Atlantic) deep convective clouds in the temperature regime where the influence of primary production is amplified by secondary ice production [Hawker *et al.*, 2021a; Hawker *et al.*, 480 2021b; Heymsfield and Willis, 2014].

Overall, we conclude that parameterisations based on samples of African dust collected from the surface and later examined in a laboratory tend to over predict the ice-nucleating activity of African dust in the atmosphere over Barbados (and possibly elsewhere). The reason for this maybe that the specific sources of dust that these parameterisations are based on may not be representative of all African dust sources. We also note that there is significant variability of the ice-nucleating activity of 485 airborne African desert dust measured in various locations downwind of the desert sources. Our hypothesis is that these differences are primarily driven by variability in K-feldspar content, which varies between ~1 and 10%. The variability in K-feldspar content and the discrepancy with laboratory derived parameterisations needs to be considered when inferring INP concentrations from mineral dust tracers in models [Schill *et al.*, 2020; Zhao *et al.*, 2021] as well as measured dust surface areas derived from lidar [Haarig *et al.*, 2019], or electron microscopy [Hande *et al.*, 2015; Sanchez-Marroquin *et al.*, 2021].

The results presented here also indicate that accurate model simulations of INP concentrations require the K-feldspar content to be tracked and accurately related to the size-resolved mineralogy of dust in the source regions. Clearly, more work is needed to improve our quantitative understanding of the ice-nucleating activity of atmospheric mineral dust. This better understanding should include a study of the ice-nucleating activity of dust from more key source regions in Africa (and elsewhere), a better understanding of the size of K-feldspar particles in source regions, and better representation of the key emission and transport 495 pathways of dust in models.





**Data availability**

The data will be made publically available in the Leeds data repository and will be linked with a doi.

**Author contribution**

ADH led the investigation and defined the project methodology and analysis. BJM, JMP and ADH led the conceptualization of the project. ADH, DOS, MPA, GCEP, CB and SW conducted the ice nucleating particle experiments during the field campaign. MDT helped with campaign planning and provided Leeds based support for the field campaign. ADH and BJM led the writing of the paper and all authors contributed to the review and editing of the paper. JBM provided supervision for ADH. EB, RC-L, AS, PS provided local knowledge, logistical support and helped in the planning and conceptualization of the project.
RH and KSC led the modelling component. CG helped with the interpretation of the data. LN performed the X-ray diffraction analysis. AS-M performed the electron microscopy analysis. OOK, MLP, CP and UP made the fine mode size distribution data measurement.

**Competing interests**

Some authors are members of the editorial board of Atmospheric Chemistry and Physics. The peer-review process was guided
by an independent editor, and the authors have also no other competing interests to declare.

**Acknowledgments**

This work was funded by the European Research Council (MarineIce, grant no. 648661) and the Natural Environment Research Council (M-Phase, grant no. NE/T00648X/1). Gaston acknowledges the National Science Foundation for a CAREER award (AGS-1944958).

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



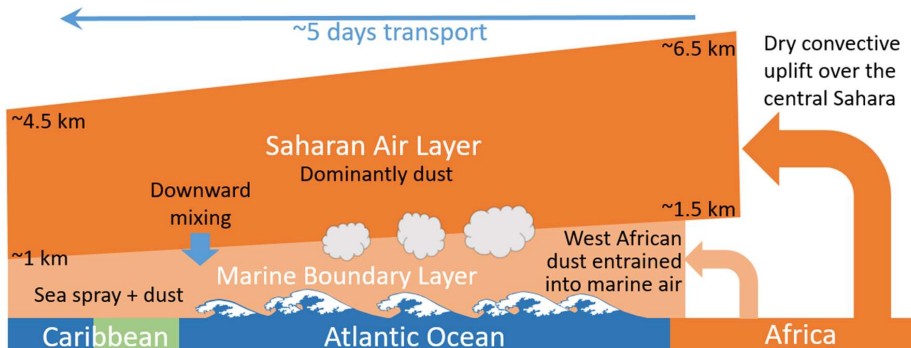


**Figure 1. Illustration of the transport of mineral dust from Africa to the summer Caribbean highlighting transport of dust through the Saharan air layer and also through the marine boundary layer.**





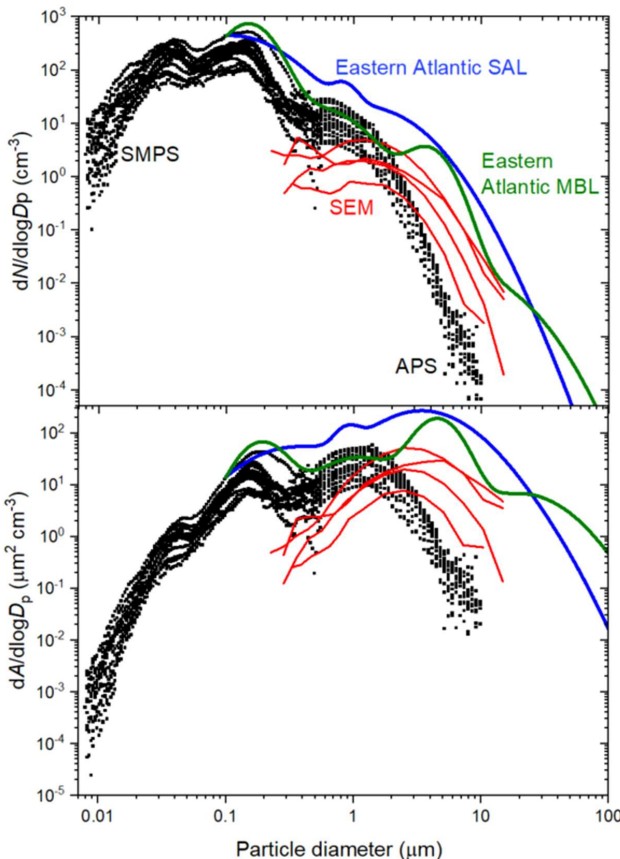

**Figure 2. Compilation of size distributions of aerosol sampled over the campaign compared to data from the eastern tropical Atlantic.**
**Data from the SMPS (scanning mobility particle sizer) and APS (aerodynamic particle sizer) spectrometer corresponding to each INP filter sample are shown (black points) alongside four distributions from the SEM (scanning electron microscope) analysis (red lines). The SMPS and APS size distributions have been merged to produce the volume equivalent diameter assuming parameters for mineral dust, see methods. Size distributions for the eastern tropical Atlantic is shown for both the SAL (blue line) and the MBL (green line) [Ryder _et al._, 2018].**


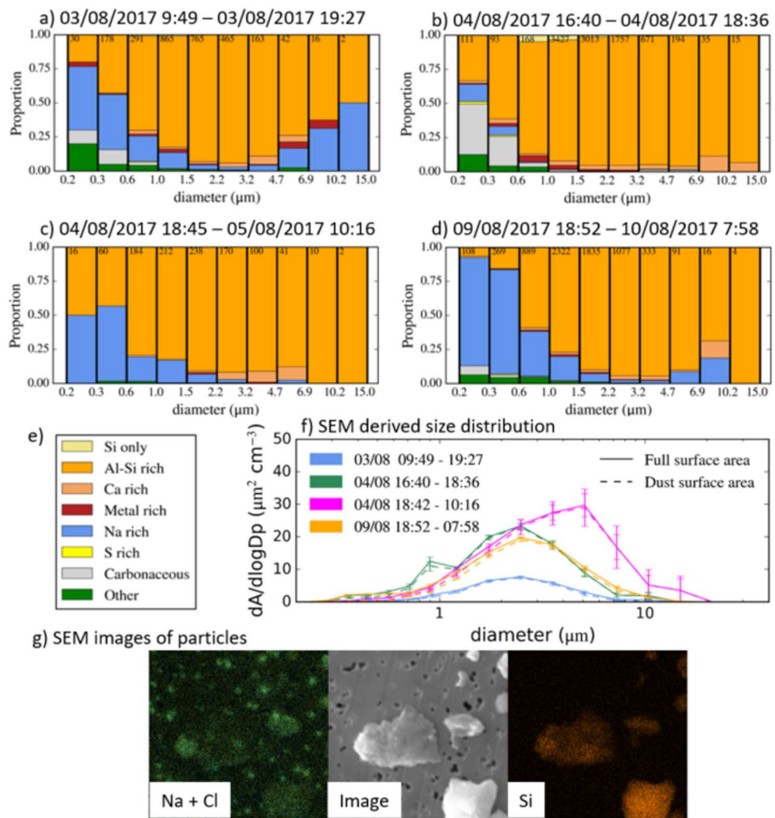

**Figure 3. Scanning electron microscope (SEM) analysis of the composition of particles collected during the campaign. Panels a to d show the size -resolved composition of the particles with number of particles per bin indicated (panel e is the key for a-d); panel f shows the surface area size distribution of all particles (solid lines) and the mineral dust category (Si, Al-Si and Ca rich; dashed lines); and panel g shows an example of several mineral dust grains in an electron microscope image (centre), the presence of Si in these (right) and particles rich in Na and Cl (i.e. fresh sea salt; left). Each image is approximately 25 µm wide.. These plots clearly show that mineral dust dominates the surface area in this location.**





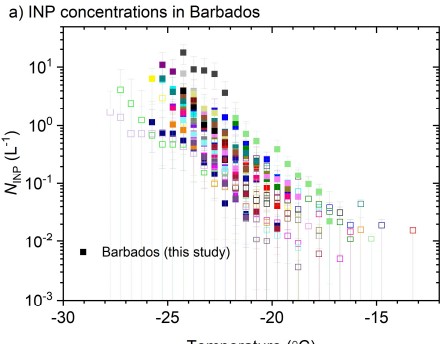

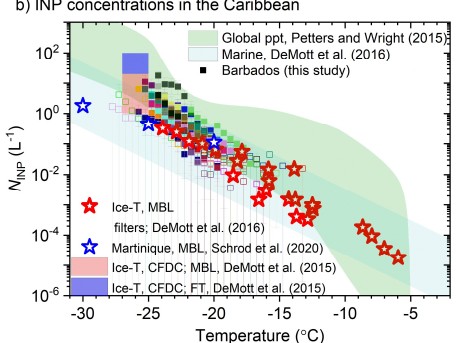

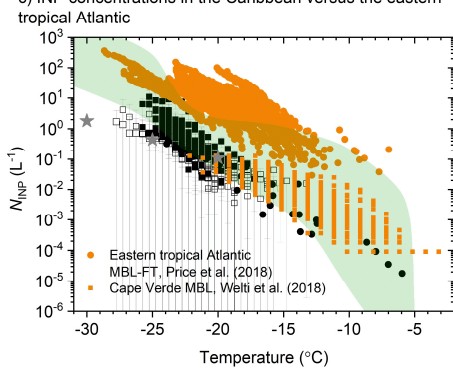

**Figure 4. INP concentrations in Barbados and the tropical Atlantic. a) Compilation of INP concentration spectra from Barbados from the 24th July-24th August 2017. See table S1 where the datapoint colour is linked to the sampling period. Data points which were consistent with the background signal are displayed as hollow points and are upper limits (see text S1). Back trajectories corresponding to the filter samples are shown in Figure S6. b) INP concentrations in the MBL of Barbados compared to other measurements in the Caribbean [DeMott et al., 2016; DeMott et al., 2015; Schrod et al., 2020] as well as the range of INP concentrations derived from precipitation samples around the globe [Petters and Wright, 2015] and a representation of INP concentrations from marine locations and wave tank experiments [DeMott et al., 2016]. The range of reported INP concentrations, at a set of temperatures, measured using a continuous flow diffusion chamber (CFDC) in the marine boundary (MBL) layer and free troposphere (FT) of the Caribbean are presented as the shaded red and blue regions (DeMott et al., 2015). c) The Caribbean INP concentrations compared with the available measurements in the eastern tropical Atlantic.**





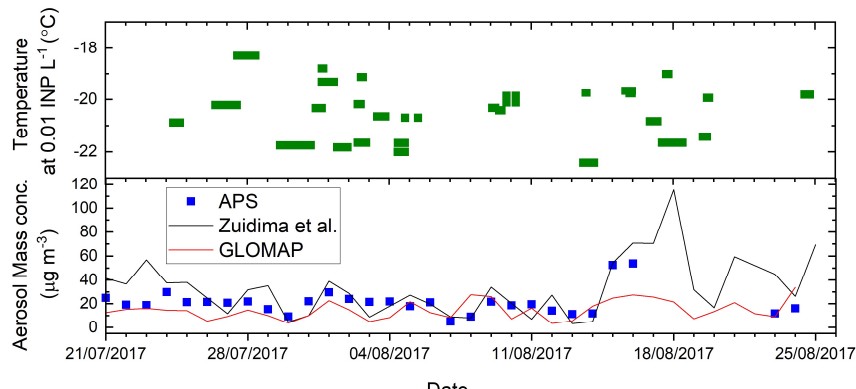

**Figure 5. Time series of the temperature at which the INP concentration was a specified value (0.01 INP L$^{-1}$) in comparison to the aerosol mass concentration derived from the APS (i.e. particles larger than 0.5 μm), based on a gravimetric analysis [Zuidema *et al.*, 2019] and predicted by a global aerosol model (GLOMAP) [Vergara-Temprado *et al.*, 2017].**




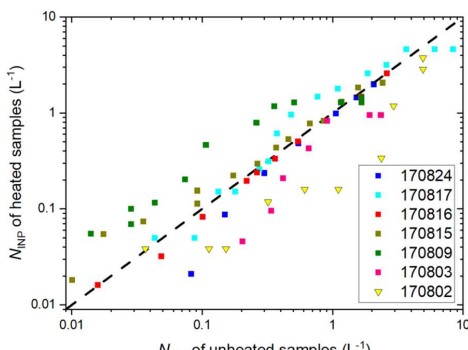

**Figure 6. Concentrations of INPs in heated versus unheated samples. See table S1 where the sampling period is indicated.  The dashed line indicates the 1:1 line.**





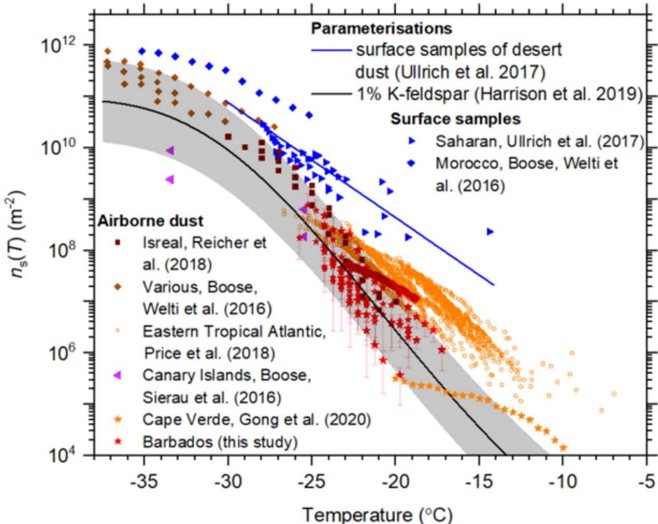


**Figure 7. Ice nucleation active site surface density, $n_s(T)$, measurements for Barbados compared with literature values for African mineral dust. We have coloured all 'surface samples' that were taken from the surface (either soil samples, or settled dust from a dust storm) and then analysed later in a laboratory with blue points [Boose _et al._, 2016b; Ullrich _et al._, 2017]. Note that we only show data from Boose et al. for the African sample that was not milled. The other literature data sets are for mineral dust sampled from**

**the air in locations in or near Africa [Boose _et al._, 2016a; Gong _et al._, 2020; Price _et al._, 2018; Reicher _et al._, 2018]. We also show the parameterisation for mineral dust samples from the surface (soil samples and dust precipitated from dust storms) from multiple arid locations around the world [Ullrich _et al._, 2017] and an $n_s(T)$ parameterisation for K-feldspar from Harrison _et al._ [2019] scaled to 1% where the grey band is the variability in the ice-nucleating activity of K-feldspar samples.**






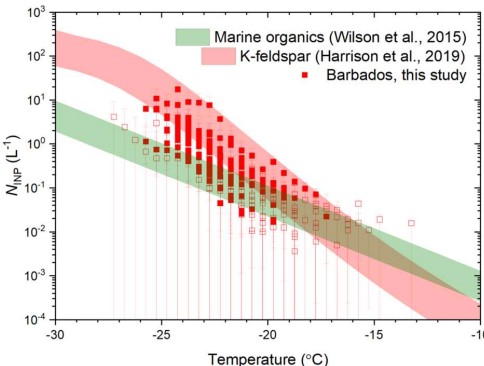

**Figure 8. Ice-nucleating particle concentrations in Barbados compared with model predictions for marine organics [Wilson *et al.*, 2015] and the K-feldspar component of desert dust [Harrison *et al.*, 2019] for the campaign period. The model simulations were done with GLOMAP and are described in Vergara-Temprado *et al.* [2017], with the exception that the parameterisation for K-feldspar from Atkinson *et al.* [2013] was replaced with that from Harrison *et al.* [2019]. Open symbols indicate upper limits. The ranges indicate the variability through the campaign period.**