# Peer review of "The ice-nucleating activity of African mineral dust in the Caribbean boundary layer"

_Atmospheric Chemistry and Physics, 2022_

## Referee Comment (RC1)

**Anonymous review of „The ice-nucleating activity of African mineral dust in the Caribbean boundary layer" by Harrison et al. (2022)**

The manuscript by Harrison et al. (2020) presents field INP (ice-nucleating particle) measurements from Barbados in the Caribbean during a campaign of one month in summer 2017. Barbados lies within the Marine Boundary Layer (MBL), but as the sampling location at Ragged Point is elevated, direct influence of sea salt is reduced. On the other hand, mineral dust is frequently transported over long distances from northern Africa and the Sahara to the Caribbean (especially in summer) and heavily influences the local aerosol composition and concentration, and therefore the ice nucleation activity. The authors confirm a rather constant presence of mineral dust during the campaign by their APS and SMPS measurements, as well as scanning electron microscopy (SEM) and powder X-ray diffraction analysis (XRD). However, the observed INP concentration and $n_s$ active site density are on the lower side, which is a little counterintuitive to what is expected at first glance. The authors list the observed low K-Feldspar content (0.1 to 2% of mineral dust mass) as the main reason for this comparably low INP activity, as K-Feldspar is known to be the driver of the INP activity of mineral dust. They then argue in detail why this lower activity was observed, namely a) the likely source region in West Africa having a lower Feldspar concentration, b) distinctly different transport patterns for the MBL and the Saharan Air Layer (SAL), c) physical and d) chemical processes during the transport, e) and the influence of the mixing of sea salt. These findings suggest that we should be more careful when applying INP parametrizations that are based on specific mineralogical compositions and size distributions and that a better understanding is needed for the K-Feldspar content around the world as well as the specific transport mechanisms and pathways.

Overall, the manuscript is well-written and follows a good structure. The presented analysis and figures are generally of good quality (except for some points which are listed below). I enjoyed reading the manuscript and I think it is definitely very relevant for the ice nucleation community, and fits the scope of the journal. In summary, I recommend to publish this interesting manuscript in ACP after minor revisions and after the following questions / comments have been addressed:

**General comment:**

The low K-Feldspar content is shown to be the most likely reason for the disparity between the observed $n_s$ activity and parametrizations (and I agree). One of the main reasons for the low K-Feldspar contribution is then given by the transport route patterns and the potential source region of the dust in West Africa (Sahel in Mauritania and Mali). The transport of mineral dust to the Caribbean and its key features are even specifically introduced in detail in section 2. However, I wonder why the actually back trajectory analysis is then cut very short in the manuscript. From the way the arguments are presented, I expected a robust trajectory and source region analysis to confirm these hypotheses. Instead the individual back trajectories are only shown in the supplement and it is only stated in one sentence that they generally support the presented idea. I think a detailed analysis would strengthen the manuscript and may drive the point home completely.

**Specific comments:**

1. Fig. 1: Maybe make the background color green for the whole "Caribbean"? I think it looks a little weird as it is, or is this somehow intended?

2. P5 L138: Is the name "IcePod" an abbreviation?

3. P8 L238: It is unclear to me which size distributions (SMPS+APS or SEM) were finally used to calculate $n_s$. Please specify. Edit: Reading the results, I now see that the APS and SMPS were used, but it is probably best to give this information in the method section as well.

4. Fig. 3: I find it rather unusual to add a legend as a separate panel to a figure. I would suggest to add the legend below the panels a-d in a line-like format or to the right of these panels.

   Further, the numbers on each size bin composition bar are very difficult to read. Maybe you can also increase the width of the figure for better readability?

   Also on the second to last sentence of the caption there are two .. at the end.

   Moreover, as the last sentence is more interpretation of the results (however obvious it may be) than a description of what is seen in the figure, I would probably leave it out of the caption.

5. P9 L276: From Fig. 5 it seems that this concentration of 100 µg m$^{-3}$ was not measured by the APS for some reason, but only by another method that is not described here but in another paper. I think this should probably be mentioned here. Especially, since the GLOMAP model line in Fig. 5 does not seem to see this peak in aerosol mass. Accordingly, it is maybe even unnecessary (?) trying to explain in detail why no INP concentration increase was observed when the aerosol data is partly missing.

6. P9 L278: On what observation is this statement about the excepted shift of 2°C per factor of 5 in surface area based? (Fig. 7? Literature?)

   Also where does this factor of 5 in surface area come from? At this point aerosol mass concentrations were discussed only.

7. Fig. 4: Shouldn't the line about the back trajectories and Fig. S6 be in the main text instead of the caption of a figure? I think the back trajectories are a key element for the story of the manuscript (see general comment).

   Panel c: Black, white and grey symbols are not indicated in the legend (although it is clear what they represent from looking at panel b).

8. Fig. 5a: It is unclear to me why this unusual INP "concentration" time series of the temperature at which the concentration was 0.01 INP L$^{-1}$ was chosen. Is this somehow considered a critical threshold for Caribbean clouds? Also looking at Fig. 4a 0.01 L$^{-1}$ seems to be a concentration at which all data points are close to or at the background concentration, if I understand the open symbols correctly. Why not choose a higher concentration (i.e., 0.1 L$^{-1}$) to circumvent this issue? Why not simply show the concentration at a given temperature (e.g., -20 °C, etc.)?

   Furthermore, I am having trouble understanding the green symbols: I guess the length of the vertical bar indicates the sampling time, correct? Further, sometimes there are up to three data points at one specific time, is this correct? It might be worthwhile to intercompare these samples in some more detail? It seems from this figure that there we some differences up to ~3 °C for this depicted concentration of 0.01 L$^{-1}$. How does this translate in variability of the INP concentration at one specific temperature?

9. Fig. 6: Some data points seem to indicate an increase in INP concentration after the heat treatment. Do you have any ideas what may have caused this? I wonder if this is a real effect caused by contamination or something else, or just variability in the measurements. If the latter were true then the language may need to be toned down a little when explaining the

differences in the text for the samples when the heat treatment did decrease the INP activity. (Although the explanations are plausible.)

Furthermore, what do the triangles mean in the figure? (Is it just to indicate the heat labile sample? If so, I wonder why the other mentioned sample is not also shown as a triangle.)

10. P10 L301 / Fig. 4c: Welti et al. (2018) present their data in detailed form for temperatures down to -16°C. Looking at the frequency distribution of measured INP concentrations at -16 °C (Fig. 5 of Welti et al., 2018) it can be seen that concentrations were usually lower than 0.1 $L^{-1}$. Considering this, to me it seems that extending the data from Welti et al. (2018) would result in a reasonably well comparable concentration range to what was measured in this manuscript. Maybe the generally higher concentrations from Price et al. (2018) are rather caused by the different altitude and possibly higher dust load, different dust composition or different particle size distribution at that height resulting from different transport patterns (or simply temporal variability)?

11. P12 L362: I think this paragraph is really relevant to explain the observations. I agree to the presented reasoning. However, I wonder if these arguments could be strengthened by providing a figure in the manuscript itself. For example, maybe showing a back trajectory frequency distribution from this campaign (or for summer months in general) superimposed on a mineralogy map (as described in the text) to really illustrate the point. Generally, I wondered why the information from the back trajectories were not featured more prominently here (there isn't even a mention of the Fig. S6 here), as they would be a key indicator that the hypothesis for the lower INP mineral dust activity were sound. Also maybe you can elaborate some more why you have low confidence in the back trajectories?

12. P14 L429 / Fig. 8: I wonder why you don't show the K-Feldspar-content "corrected" GLOMAP prediction in the figure (e.g., as an average line).

13. Fig. S4: I am not familiar with this method. Could you briefly describe it (here or in the manuscript) and explain what this figure means? What are negative counts? How do you estimate the relative mineral content? What is the grey and blue line? What is the x-axis?

14. Fig. S9: Is this mass fraction calculated for the surface or a mean column density?

**Technical comments:**

15. References should be put in parentheses ( ) instead of square brackets [ ] according to the journal guidelines

16. P3 L74: Missing space between "sources. In"

17. P4 L97: "rides above" – I am not sure about this phrasing. Maybe rephrase?

18. P6 L162: → section **3**.3

19. P6 L176: → M**ö**hler et al (2008)

20. P7 L191: what do you mean by "where the square brackets indicate the concentration"? Also there is a right parenthesis but no left one.

21. P9 L277 / Fig. 5: There is no explicitly labeled panel a) and b) in Fig. 5 yet.

22. P11 L331: SAL layer → SAL has the word layer already in it

23. P11 L337: The title of chapter 6 is rather long. Maybe it can be rephrased to a more concise title?

24. P14 L409: → Vergara-Temprado et al. (2017)

25. P14 L426: → Fig. S9

26. Fig. S5: The error bars reach out of the figure. Also, there is no mention of Fig. S5 in the manuscript yet.

27. Fig. S6: There is a weird line next to the caption and red underscore symbols after "2000" and "4000" in the caption.

28. Fig. S7: Panels e and f are already in the manuscript and do not need to be shown here again. The panels are not correctly assigned in the caption.

**References:**

Price, H. C., et al. (2018), Atmospheric Ice-Nucleating Particles in the Dusty Tropical Atlantic, J. Geophys. Res., 123(4), 2175-2193, doi:10.1002/2017JD027560.

Welti, A., K. Müller, Z. L. Fleming, and F. Stratmann (2018), Concentration and variability of ice nuclei in the subtropical maritime boundary layer, Atmos. Chem. Phys., 18(8), 5307-5320, doi:10.5194/acp-18-5307-2018.

---

## Author Comment (AC1)

We thank the referee for their comments.  Our responses are in blue text.

**General comment:**

*The low K-Feldspar content is shown to be the most likely reason for the disparity between the observed $n_s$ activity and parametrizations (and I agree). One of the main reasons for the low K-Feldspar contribution is then given by the transport route patterns and the potential source region of the dust in West Africa (Sahel in Mauritania and Mali). The transport of mineral dust to the Caribbean and its key features are even specifically introduced in detail in section 2. However, I wonder why the actually back trajectory analysis is then cut very short in the manuscript. From the way the arguments are presented, I expected a robust trajectory and source region analysis to confirm these hypotheses. Instead the individual back trajectories are only shown in the supplement and it is only stated in one sentence that they generally support the presented idea. I think a detailed analysis would strengthen the manuscript and may drive the point home completely.*

The short answer is that back trajectories cannot readily be used to infer source regions in Africa. Trajectories for the Caribbean MBL typically do not intersect with Africa (see the figures in the SI, but also the one included below), yet they systematically contain dust.  So, the dust must enter the MBL at some point between Africa and the Caribbean through processes that are not resolved in a simple back trajectory analysis. Hence, it is not possible (or at least very challenging) to use such an analysis to say anything about African source regions.  This is why we relied on other evidence from the literature on source regions of dust found in the Caribbean.

[Figure]

Figure 1. Back trajectory frequency plot for 200 m above Ragged Point.  The trajectories are run for 9 days.  The period is 10 years, run across all seasons.

We have updated the pertinent paragraph in section 6.1 to make it clear why we didn't do more with the back trajectories:

"We examined back trajectories corresponding to our samples and while they are consistent with the fact that there was generally a westerly flow and that the SAL is derived from the African continent, trajectories do not allow us to resolve African source regions. As discussed in section 2, the transport of dust from African source regions to the Caribbean MBL is complex and poorly represented in models. Trajectories relevant for the Caribbean MBL often do not intercept the African coast; instead these trajectories often have a strong northerly component (see SI Figure S6). Hence, dust is not transported directly from source to the Caribbean, instead processes such as the mixing of dust down from the SAL into the MBL in moist convective erosion at the base of the SAL somewhere between Africa and the Caribbean must take place (Reid et al., 2003; Prospero and Carlson, 1972). Hence, the trajectory models cannot be used to indicate specific sources of African dust."

*Specific comments:*
*1. Fig. 1: Maybe make the background color green for the whole "Caribbean"? I think it looks a little weird as it is, or is this somehow intended?*

We intended this to indicate islands, but have changed it to solid green.

*2. P5 L138: Is the name "IcePod" an abbreviation?*

No. It is a name.

*3. P8 L238: It is unclear to me which size distributions (SMPS+APS or SEM) were finally used to calculate $n_s$. Please specify. Edit: Reading the results, I now see that the APS and SMPS were used, but it is probably best to give this information in the method section as well.*

This has now been made very clear

*4. Fig. 3: I find it rather unusual to add a legend as a separate panel to a figure. I would suggest to add the legend below the panels a-d in a line-like format or to the right of these panels.*

Changed

*Further, the numbers on each size bin composition bar are very difficult to read. Maybe you can also increase the width of the figure for better readability?*

We will supply a vector graphic file for final typesetting where this will be clearer.

*Also on the second to last sentence of the caption there are two .. at the end.*

Corrected

*Moreover, as the last sentence is more interpretation of the results (however obvious it may be) than a description of what is seen in the figure, I would probably leave it out of the caption.*

Removed

*5. P9 L276: From Fig. 5 it seems that this concentration of 100 μg m$_{-3}$ was not measured by the APS for some reason, but only by another method that is not described here but in another paper. I think this should probably be mentioned here. Especially, since the GLOMAP model line in Fig. 5 does not seem to see this peak in aerosol mass. Accordingly, it is maybe even unnecessary (?) trying to explain in detail why no INP concentration increase was observed when the aerosol data is partly missing.*

This was a minor point and as the referee says, our data was unfortunately incomplete at this time, so we have removed the sentences

*6. P9 L278: On what observation is this statement about the excepted shift of 2°C per factor of 5 in surface area based? (Fig. 7? Literature?)*

This was based on the K-feldspar parameterisation. But, we have removed these lines.

*Also where does this factor of 5 in surface area come from? At this point aerosol mass concentrations were discussed only.*

We assumed that the surface area would scale with the mass, which is obviously a crude assumption. We have removed these lines.

*7. Fig. 4: Shouldn't the line about the back trajectories and Fig. S6 be in the main text instead of the caption of a figure? I think the back trajectories are a key element for the story of the manuscript (see general comment).*

Now referred to in the text.

*Panel c: Black, white and grey symbols are not indicated in the legend (although it is clear what they represent from looking at panel b).*

We didn't include an item in the key for panel c because as the referee says it is clear what they are from panel b.

*8. Fig. 5a: It is unclear to me why this unusual INP "concentration" time series of the temperature at which the concentration was 0.01 INP $L^{-1}$ was chosen. Is this somehow considered a critical threshold for Caribbean clouds? Also looking at Fig. 4a 0.01 $L^{-1}$ seems to be a concentration at which all data points are close to or at the background concentration, if I understand the open symbols correctly. Why not choose a higher concentration (i.e., 0.1 $L^{-1}$) to circumvent this issue? Why not simply show the concentration at a given temperature (e.g., -20 °C, etc.)?*

There was an error here. This should be the INP concentration at 0.1 $L^{-1}$. We have corrected the main paper and the SI.

The reason we have plotted this quantity rather than a concentration at a particular T is twofold: i) because our data sits roughly between an upper and lower limit in INP conc and shifts to higher or lower temperatures depending on activity it is not always possible to plot a temperature time series because we cannot always report data at a single T in all runs (we might be able to do this for this dataset at around -23 C). ii) What is more relevant, the concentration at some arbitrary T or the T at which this concentration is reached? I would argue that for clouds it is the latter. We picked 0.1 $L^{-1}$ because it is well above our baseline and is a concentration where we expect to start to see substantial changes to a cloud. We recently used this in Porter et al. (2022)

*Furthermore, I am having trouble understanding the green symbols: I guess the length of the vertical bar indicates the sampling time, correct?*

Yes, correct. This has been made clear in the caption.

*Further, sometimes there are up to three data points at one specific time, is this correct? It might be worthwhile to intercompare these samples in some more detail? It seems from this figure that there we some differences up to ~3 °C for this depicted concentration of 0.01 $L^{-1}$. How does this translate in variability of the INP concentration at one specific temperature?*

Correct, there was some overlap. We had some technical problems with our samplers (they cut out at random times), which prevented us from sampling for set durations of time. Our

original intention was to have two filter samples in parallel, one for INP and one for SEM or a spare for INP analysis.  But, this wasn't possible, hence the sampling durations varied considerably. Because of the erratic sampling we didn't have many cases where we had samples taken at exactly the same time. The quoted temperature uncertainty of the instrument is 0.4°C, hence the differences appear to be significant. The only time we had almost identical sampling was on the 4[th] August and the results of the two were well within the quoted uncertainty. We have added the following text to section 5.1 where we had removed text based on the referee's other suggestions:

"Figure 5 illustrates the time series of the temperature at which the INP concentration was 0.1 L-1, again showing that the INP spectra were relatively invariant through the campaign. For technical reasons we performed very few parallel samplings where the same volume of air was sampled over the same period of time on multiple filters (we had technical problems with our samplers).  However, on the 4th August we did manage to do this and found excellent reproducibility between the two samples, with the temperature at 0.1 INP L-1 being within the 0.4°C instrument uncertainty (Whale et al., 2015). The variability in the INP spectra of ~3 K would therefore appear to be related to the activity of the aerosol. In addition, the aerosol (and dust) mass concentrations were also relatively constant during the campaign (Figure 5), as were the meteorological parameters of pressure, relative humidity and temperature (see Figure S7). The variability in dust mass concentration in our campaign in 2017 was substantially less than that in other years (Zuidema et al., 2019), hence a campaign in a different year may have found more variability in INP concentrations."

*9. Fig. 6: Some data points seem to indicate an increase in INP concentration after the heat treatment. Do you have any ideas what may have caused this? I wonder if this is a real effect caused by contamination or something else, or just variability in the measurements. If the latter were true then the language may need to be toned down a little when explaining the differences in the text for the samples when the heat treatment did decrease the INP activity. (Although the explanations are plausible.)*

In our recent paper on characterising the heat tests (Daily et al., 2022) we started plotting heat test data using box and whisker plots.  This is much better than the 1:1 plot that we used here.  We have updated the figure and this gives a much clearer picture of what the heat tests show.  Yes, in a few samples there was a marginally higher freezing T, but this was on the order of our experimental uncertainty.

*Furthermore, what do the triangles mean in the figure? (Is it just to indicate the heat labile sample? If so, I wonder why the other mentioned sample is not also shown as a triangle.)*

There was no significance.

*10. P10 L301 / Fig. 4c: Welti et al. (2018) present their data in detailed form for temperatures down to -16°C. Looking at the frequency distribution of measured INP concentrations at -16 °C (Fig. 5 of Welti et al., 2018) it can be seen that concentrations were usually lower than 0.1 L-1. Considering this, to me it seems that extending the data from Welti et al. (2018) would result in a reasonably well comparable concentration range to what was measured in this manuscript. Maybe the generally higher concentrations from Price et al. (2018) are rather caused by the different altitude and possibly higher dust load, different dust composition or different particle size distribution at that height resulting from different transport patterns (or simply temporal variability)?*

We have removed the reference to -15 C and now state: "hence the actual INP concentrations may have extended to greater values at the lower end of their temperature range".

*11. P12 L362: I think this paragraph is really relevant to explain the observations. I agree to the presented reasoning. However, I wonder if these arguments could be strengthened by providing a figure in the manuscript itself. For example, maybe showing a back trajectory frequency distribution from this campaign (or for summer months in general) superimposed on a mineralogy map (as described in the text) to really illustrate the point. Generally, I wondered why the information from the back trajectories were not featured more prominently here (there isn't even a mention of the Fig. S6 here), as they would be a key indicator that the hypothesis for the lower INP mineral dust activity were sound. Also maybe you can elaborate some more why you have low confidence in the back trajectories?*

We have addressed this in response to the 'general point' above.

*12. P14 L429 / Fig. 8: I wonder why you don't show the K-Feldspar-content "corrected" GLOMAP prediction in the figure (e.g., as an average line).*

We had this in an earlier version and it caused confusion, hence we decided to leave it out.

*13. Fig. S4: I am not familiar with this method. Could you briefly describe it (here or in the manuscript) and explain what this figure means? What are negative counts? How do you estimate the relative mineral content? What is the grey and blue line? What is the x-axis?*

Fig S4 has been improved, with better axis labelling. We have also expanded the caption to give some brief indication of how to interpret the plot. The caption now reads:

'**Figure S4**: Powder X-ray diffraction analysis and Rietveld refinement of aerosol collected in rain water from the 3$^{rd}$-4$^{th}$ of August 2017. The proportion of each mineral identified in the sample is shown in the key. The limit of detection of this technique was ~2 wt%, hence K-feldspar was below this limit, thus defining an upper limit. X-ray diffraction combined with Rietveld refinement analysis is a technique for quantifying the proportions of the crystalline components of a sample. Crystalline components have a set of sharp Bragg peaks that are characteristic of a particular crystalline material. The X-ray diffractometer (Bruker D8) was equipped with a source supplying X-rays at 1.54060 Å. The diffraction pattern is shown in blue, while the fitted Rietveld pattern (a combination of the patterns associated with a range of minerals) is shown in pink. The positions of the Bragg peaks are associated with each mineral are shown below the patterns and are colour coded according to the key. The grey line is the residual (i.e. the difference between the fitted and measured patterns.'

*14. Fig. S9: Is this mass fraction calculated for the surface or a mean column density?*

This is at the surface. This has been added to the caption.

**Technical comments:**
*15. References should be put in parentheses ( ) instead of square brackets [ ] according to the journal guidelines*

Corrected

*16. P3 L74: Missing space between "sources. In"*

Corrected

*17. P4 L97: "rides above" – I am not sure about this phrasing. Maybe rephrase?*

Changed to 'resides'

*18. P6 L162: → section **3**.3*

Corrected

*19. P6 L176: → Möhler et al (2008)*

Corrected

*20. P7 L191: what do you mean by "where the square brackets indicate the concentration"? Also there is a right parenthesis but no left one.*

This has been corrected. We have changed the way we represent concentration to be consistent with Copernicus rules.

*21. P9 L277 / Fig. 5: There is no explicitly labeled panel a) and b) in Fig. 5 yet.*

Corrected

*22. P11 L331: SAL layer → SAL has the word layer already in it*

Corrected

*23. P11 L337: The title of chapter 6 is rather long. Maybe it can be rephrased to a more concise title?*

Shortened to 'Why African dust in the MBL in Barbados has a relatively low ice-nucleating activity'

*24. P14 L409: → Vergara-Temprado et al. (2017)*

Corrected

*25. P14 L426: → Fig. S9*

Corrected

*26. Fig. S5: The error bars reach out of the figure. Also, there is no mention of Fig. S5 in the manuscript yet.*

Corrected the figure.

We have added '(in Figure S5 we also show the predicted INP concentration associated with plagioclase, albite and quartz as well as K-feldspar, showing that K-feldspar contributes more to the ice-nucleating activity of dust in this location than the other minerals)'

*27. Fig. S6: There is a weird line next to the caption and red underscore symbols after "2000" and "4000" in the caption.*

Corrected

*28. Fig. S7: Panels e and f are already in the manuscript and do not need to be shown here again. The panels are not correctly assigned in the caption.*

We feel it is important to include all panels in order that we can compare them. Caption corrected.

**References**

Daily, M. I., Tarn, M. D., Whale, T. F., and Murray, B. J.: An evaluation of the heat test for the ice-nucleating ability of minerals and biological material, Atmos. Meas. Tech., 15, 2635-2665, 10.5194/amt-15-2635-2022, 2022.

Porter, G. C. E., Adams, M. P., Brooks, I. M., Ickes, L., Karlsson, L., Leck, C., Salter, M. E., Schmale, J., Siegel, K., Sikora, S. N. F., Tarn, M. D., Vüllers, J., Wernli, H., Zieger, P., Zinke, J., and Murray, B. J.: Highly Active Ice-Nucleating Particles at the Summer North Pole, Journal of Geophysical Research: Atmospheres, 127, e2021JD036059, https://doi.org/10.1029/2021JD036059, 2022.

Prospero, J. M. and Carlson, T. N.: VERTICAL AND AREAL DISTRIBUTION OF SAHARAN DUST OVER WESTERN EQUATORIAL NORTH-ATLANTIC OCEAN, Journal of Geophysical Research, 77, 5255-&, 10.1029/JC077i027p05255, 1972.

Reid, J. S., Kinney, J. E., Westphal, D. L., Holben, B. N., Welton, E. J., Tsay, S. C., Eleuterio, D. P., Campbell, J. R., Christopher, S. A., Colarco, P. R., Jonsson, H. H., Livingston, J. M., Maring, H. B., Meier, M. L., Pilewskie, P., Prospero, J. M., Reid, E. A., Remer, L. A., Russell, P. B., Savoie, D. L., Smirnov, A., and Tanre, D.: Analysis of measurements of Saharan dust by airborne and ground-based remote sensing methods during the Puerto Rico Dust Experiment (PRIDE), Journal of Geophysical Research-Atmospheres, 108, 10.1029/2002jd002493, 2003.

Whale, T. F., Murray, B. J., O'Sullivan, D., Wilson, T. W., Umo, N. S., Baustian, K. J., Atkinson, J. D., Workneh, D. A., and Morris, G. J.: A technique for quantifying heterogeneous ice nucleation in microlitre supercooled water droplets, Atmos. Meas. Tech., 8, 2437-2447, 10.5194/amt-8-2437-2015, 2015.

Zuidema, P., Alvarez, C., Kramer, S. J., Custals, L., Izaguirre, M., Sealy, P., Prospero, J. M., and Blades, E.: Is Summer African Dust Arriving Earlier to Barbados? The Updated Long-Term In Situ Dust Mass Concentration Time Series from Ragged Point, Barbados, and Miami, Florida, Bulletin of the American Meteorological Society, 100, 1981-1986, 10.1175/bams-d-18-0083.1, 2019.

---

## Author Comment (AC2)

**Response to referee 2**

We thank the referee for their comments. Our responses are in blue text.

*A - The main message of the manuscript is that a low INP acitivity of the MLB aerosol particles is detected in Barbados and this is most likely due to the low K-feldspar content. I accept this thesis, but find that even more convincing data could possibly have been obtained from the samples taken. Above all, it should be clearly shown that the K feldspar was indeed very low throughout the campaign.*
*For this purpose, a more precise determination of the real K-feldspar content within the individual samples would be desirable. The manuscript mentions XRD measurements, but only one measurement of a rain sample is presented. Were the daily samples also examined with XRD and if so, what was the variability of the K-feldspar values?*

We agree that measurement of K-feldspar content throughout the campaign would have been valuable and if we were to go back and do it again we would have a much stronger focus on the mineralogy. However, measuring mineralogy of airborne dust is not trivial and would require significant investment in higher volume samplers and in the X-ray diffraction technique development. The rain water sample was a one off and opportunistic, where a sample of rain water from the Mesa lab sampler trap happed to have a lot of dust in it, allowing for this XRD analysis. Even this quantity of dust proved challenging to work with. But, this analysis alongside the SEM and literature work all clearly point to a relatively low K-feldspar content.

*Otherwise, the K-feldspar content is derived from the scanning electron microscopic data. This is possible in principle, but it requires several things. On the one hand a very low particle assignment of the filter, so that a single particle analysis remains possible and on the other hand the analysis of at least about 100 particles / analyzed size bin. Figure 3 shows the scanning electron microscopic results of 4 samples indicating significant variability (at least with regard to sea salt). Contrary to the figure legend, only percentage values can be seen, not a number of analyzed particles, which would be desirable. I would have liked to support the assumption of the low K-feldspar content in general throughout the campaign either by the XRD data from more samples or electron microscopic data.*

The number of particles per bin is already given. These numbers should be more obvious with the higher resolution image that will be supplied for the typesetting.

As mentioned above, we would also have like to have done more work on the mineralogy. We mentioned the limitations of the X-ray diffraction analysis above, but we also unfortunately suffered from problems with our Meslab filter samplers. These samplers cut out at random times and while our plan was to always have at least two concurrent samples, one for INP analysis and one for SEM analysis, this was unfortunately not possible.

*B – The discussion on other factors, influencing IN activity (based on literature/comparative values), is generally very thorough and broad. In some cases, there are even repetitions here, which could be avoided by some cuts.*
*I am not entirely convinced by the discussion about the assumed low influence of internal mixing with sea salt.*

We left this somewhat open because we cannot rule it out completely. More work should be done on the role of sea salt in the ice nucleating ability of dust in the future.

*In the 4 compositions shown in Figure 3, a larger proportion of external sea salt particles in coarse mode is recognizable in 2 samples. However, the extent to which internal mixing with sea salt was found in the "Al-Si-rich" particles (should be visible in SEM) is not used*

*for the discussion. Is there SEM data for more samples? This would also help to get an
idea of the variabilty of aerosol particle composition. These data could significantly strengthen the
discussion and conclusions.*

There is certainly a degree of internal mixing. We already mentioned this in section 4 where we state 'The mineral dust was often internally mixed with sea salt, similar to a previous study of aerosol at Ragged Point (Kandler et al., 2018), and sea salt (Na rich category) also appeared as an externally mixed (mostly submicron) particle type.'.  We have also added 'Given mineral dust particles are internally and externally mixed with sea salt (Figure 3), we now consider if the salt may have altered the nucleating ability of the dust.' To the beginning of section 6.2.

As discussed above, we had limited number of samples for SEM analysis.

*Minor points:
A - Figure 2 shows size distributions. In red those that were determined in the SEM. Size
distribution determinations with the help of SEM are difficult and a factor of 2 as an error
and a flatter course is quite typical in this type of determination.
The size distribution from SMPS/APS should still have the higher accuracy. Therefore, it
would be advisable not to use the size distribution determined with the help of the SEM
data but to combine the relative group abundance (SEM data) with the SMPS/APS size
distribution in order to achieve the most reliable results.*

For the surface areas we used to derive active site densities, we used the APS/SMPS data.  Since we only had four SEM size distributions for the reasons defined above, we cannot use the SEM data to routinely come up with a combined size distribution.

*B- Line 72 Reischel, 1987 used ammonium iodide. These results are difficult to compare
with atmospheric IN measurements.*

This reference is cited because it was the first, to our knowledge, to point out that there a solute dependence involving ammonium ions that can enhance nucleation.

**References**

Kandler, K., Schneiders, K., Ebert, M., Hartmann, M., Weinbruch, S., Prass, M., and Pöhlker, C.: Composition and mixing state of atmospheric aerosols determined by electron microscopy: method development and application to aged Saharan dust deposition in the Caribbean boundary layer, Atmos. Chem. Phys., 18, 13429-13455, 10.5194/acp-18-13429-2018, 2018.